# Intensity Characteristics and Multi-Scenario Projection of Land Use and Land Cover Change in Hengyang, China

**DOI:** 10.3390/ijerph19148491

**Published:** 2022-07-12

**Authors:** Zhiwei Deng, Bin Quan

**Affiliations:** 1College of Geography and Tourism, Hengyang Normal University, Hengyang 421002, China; dzw17673290352@aliyun.com; 2Hengyang Base of International Centre on Space Technologies for Natural and Cultural Heritage under the Auspices of UNESCO, Hengyang 421002, China

**Keywords:** land use and land cover change, Intensity Analysis, patch-generating simulation, Hengyang, China

## Abstract

Intensity Analysis has generally been applied as a top-bottom hierarchical accounting method to understand regional dynamic characteristics of land use and land cover (LULC) change. Given the inconvenience of transition level in the detailed and overall presentation of various category transitions at multiple intervals, a novel transition pattern is proposed to represent the transition’s size and intensity and to intuitively identify the stationary mode of transition, which helps the transition level to connect to the mode with the process. Intensity Analysis was conducted to communicate the transition between LULC categories in Hengyang from 1980 to 2015. The patch-generating land use simulation (PLUS) model was employed for multi-scenario projection from 2015 to 2045. From 1980 to 2015, 2005 was a significant turning point in the speed of LULC change in Hengyang, and the change rate after this time point was three times that before the time point. The gain of built-up and bare, and the loss of cultivated was always active. The reason for the large loss of forest is that forest comprises the largest proportion of Hengyang. The loss of cultivated and the loss of forest contributing to the built-up’s gain is much larger, but the mechanism behind the transition differed. A stationary targeting transition mode from cultivated to built-up in Hengyang was detected. The PLUS model confirmed that the area of forest, cultivated and grass will reduce, and the rate of decrease will slow down in the future, while water areas will slightly increase. Our work enriches the methodology of Intensity Analysis and provides a scientific reference for the sustainable development and management of land resources in Hengyang.

## 1. Introduction

Human activities are changing the Earth’s systems in a way that threatens well-being and development [1], and they have a significant impact on the environment at the local, regional and global levels [2]. With the deepening global change research, land use and land cover (LULC) change has become the core component of global environmental change research [3]. LULC change is both the reason and result of biophysical processes and the social economy, with great influence on climate change, biodiversity, grain yield, and air contamination [4]. Measuring the dynamics of LULC change and modeling effective spatial projections are important aspects to deal with environmental change and to achieve regional sustainable development, which is considered a necessary means to better understand and address socio-economic and land resource issues [5,6].

Over the past decades, remote sensing (RS) and geographical information systems (GIS) have provided strong data sources and technical support for the monitoring and detecting of LULC change and urban expansion and sprawl [7]. Compared to conventional ground surveys, RS has remarkably helped in the efficient acquisition and storage of Earth surface data, which facilitates the quantification of long-time series LULC dynamics [8]. Currently, the common methods to quantify the temporal change of LULC maps at several points in time include single and comprehensive dynamic degree [9], transfer matrix [10], change trajectory [11], and Intensity Analysis [12]. However, the popular single dynamic degree only considers the net change of category, which is the absolute value of the gain minus loss, thus ignoring the simultaneous expansion and reduction of the category spatially. For comprehensive dynamic degree, readers have used multiple ways to calculate the degree for confusing mathematical notation [13]. Furthermore, this degree is the sum of the loss intensities of different categories, which is proven to have no practical interpretation [14]. In change trajectory, the LULC maps in various time points are overlaid by raster calculations, and are then counted for different category encodings composed of a series of numbers, which provides abundant transfer information. However, it is similar to the transfer matrix to count the size of changes between categories, neglecting the interpretation of the impact of the category’s initial size on changes. Intensity Analysis as one systematical method can characterize which processes of LULC change are intensive compared to random or uniform [12]. This method has been adopted in the study of LULC change, urban expansion [15,16], desertification [17,18], and regional comparison [19,20,21]. Stationary in Intensity Analysis refers to the transition from category i to category j that shows the same targeting or avoiding characteristics at all intervals [12]. To diagnose, the information is needed for researchers and decision makers to help governments better formulate land use policies.

However, when the original Intensity Analysis involves more categories and intervals, its transition level results are difficult to intuitively and quickly reflect the size and intensity of a variety of transition processes, as well as the stationary mode. Accordingly, the transition level was taken further through optimization and visualization by graphics expression and cross-contingency tables [22,23,24]. The improvement of this is not the integration of size, intensity, and stationary mode between the category’s transition, although it can show all transition intensities and whether each transition process targets or avoids. Scientific analysis of the stationary modes and mechanisms in regional LULC change has been an enlightenment for promoting regional sustainable development and innovating land resource utilization. Thus, further improvement in this way is always expected. Specifically, based on the transition level of the Intensity Analysis, a novel transition pattern is constructed to quickly and intuitively represent the size, intensity, and mode of land transition in this study.

To comprehend probability trends in LULC, the spatiotemporal simulation of LULC is used as a powerful tool to analyze the consequences and the underlying driving force of regional human activities interacting with the natural environment [25]. Cellular automata (CA) was first applied to geographical modeling in 1979 [26], mainly used in urban development studies at the outset [27,28], which effectively relates urban expansion to socio-economic development through the defining transition rules. However, the single model or method still was flaws despite its existing merits, hence the need for integrating modeling in LULC simulations. The coupling of the CA and Markov Chain (MC) model has become a common method to simulate LULC evolution and spatial distribution due to its prediction based on process state, strong parallel computing, and dynamic expression ability [29]. A large number of scholars have conducted in-depth research in this field and have gradually developed a series of hybrid models and algorithms, such as the DT-based urban expansion model [30], multi-criteria evaluation (MCE)-CA-MC [31], logistic regression (LR)-CA-MC [32], SLEUTH model [33], artificial neural network (ANN)-CA [34], MLP-CA-MC [35], CLUE-S model [36], FLUS model [25], etc. Recently, a patch-generating land use simulation (PLUS) model was proposed to simulate land use patch change and analyze the potential driving of LULC dynamics [37]. The PLUS model improves the previous mining methods of transition rules, such as transition analysis strategy (TAS) and pattern analysis strategy (PAS). A new land expansion analysis strategy (LEAS) is proposed on the random forest (RF) algorithm and maintains the advantages of adaptive inertia competition and roulette competition mechanism of the future land use simulation (FLUS) model [38]. Multi-type random patch seeds (CARS) are applied to model multiple LULC types at fine-scale resolution. The RF algorithm as a machine learning method with a powerful fitting capacity is suitable for mining the complicated transition rule of the CA model [39]. Many studies have shown that the PLUS model can more accurately simulate the spatial pattern and evolution process of LULC compared with other models [40,41].

China has experienced unprecedented economic growth and urbanization since the launch of economic reforms in the late 1970s, which also brings a drastic change in the spatiotemporal pattern of LULC [42]. In several study cases, many scientists have investigated the LULC of urban agglomerations and developed areas in China [43], such as Beijing–Tianjin–Hebei [44], the Yangtze River Delta [45], Guangdong–Hong Kong–Macao Greater Bay Area [46,47] and national central cities [48]. However, socioeconomic development of urban areas in different regions, different cultures, and different levels often experiences various LULC change processes. Therefore, under the background of rapid urbanization and rural revitalization in China, it is necessary to systematically understand the LULC change in medium and underdeveloped regions. As one of the old industrial bases in central China, Hengyang is playing a bridgehead in undertaking coastal industrial transfer. Meanwhile, Hengyang is one of the typical representatives of hilly ecological fragile areas in south China. In recent decades, Hengyang’s rapid economic development has been attended by a large population that has increased in concentration and urban expansion and sprawl, which result in the buildup of pressure on land and in the reduction of agricultural land and forests [49]. In the past, several scholars have studied the land change in Hengyang from the perspectives of ecological security of land [50], urban boundary delineation [51], projection under a single scenario [43], and urban expansion simulation [52]. However, the in-depth quantitative analysis of a long-term LULC change in Hengyang is still lacking.

Therefore, to analyze and model LCLC change in Hengyang, the main objectives of this study were as follows: (1) how to apply the improved Intensity Analysis framework to study the size, intensity, and stationary characteristics of regional LULC change; (2) various LULC changes in Hengyang under different scenarios. Building on high-resolution LULC observation data, the improved Intensity Analysis was employed to explore the intensity of LULC change and its stationary characteristics in Hengyang from 1980 to 2015, and then the PLUS model was applied to project the spatial pattern of LULC in Hengyang under different scenarios from 2015–2045. Our study enriches the analysis ability of the transition level of Intensity Analysis and has certain guiding significance for the sustainable planning, utilization, and protection of land in Hengyang. In addition, it provides an important reference for the land management of other medium-developed cities in central China.

## 2. Materials and Methods

### 2.1. Study Area

Hengyang is located in the south of Hunan Province and the middle reaches of the Xiangjiang River (110°32′16″~113°16′32″ E, 26°07′05″~27°28′24″ N). The terrain is high around the perimeter and low in the middle, forming the “Hengyang Basin” with a total land area of about 15,310 km^2^ (Figure 1). The study area belongs to a subtropical monsoon climate, with an annual average temperature of about 18 °C and an average annual rainfall of about 1352 mm. In recent years, the economy of Hengyang in Hunan Province has maintained rapid growth. In 2020, Hengyang achieved CNY 350.8 billion in GDP, ranking fourth in the province, and is the second-largest city with a population of 6.64 million. The urbanization rate is 54.27% [53]. Hengyang has jurisdiction over seven counties (Figure 1b).

### 2.2. Data Source and Processing

In this study, the LULC data in Hengyang in 1980, 1995, 2005, and 2015 (30 m resolution) are derived from the remote sensing monitoring dataset of land use and land cover in China, which can be provided by the Data Center for Resource and Environment Sciences, Chinese Academy of Sciences (http://www.resdc.cn, accessed on 28 June 2021). The dataset was made by human–computer interaction visual interpretation based on Landsat TM/ETM+/OLI images, and its overall accuracy is higher than 90% [54,55]. According to the analysis for the study area, the LULC category in Level I is named as cultivated, forest, grass, water, built-up, and bare. The PLUS model requires driving factors such as the social economy and natural environment data. Therefore, road networks, governments, water, elevation, and other driving factors of LULC change in Hengyang were used to analyze and simulate the dynamic evolution of land. The GDP, population grid data, annual precipitation and temperature distribution of 1 km resolution data, and soil type data were collected from the Data Center for Resource and Environment Sciences, Chinese Academy of Sciences (http://www.resdc.cn, accessed on 19 October 2021). Soil types include acidic purple soil, paddy soil, red soil, and 11 other soil types. Traffic road data, including multi-level highway networks, high-speed railways, railways, highways, and other vector sources were taken from the non-secret vector map data product (1:1 million scales). This product is provided by the National Geomatics Center of China (https://www.webmap.cn, accessed on 28 October 2021). The primary road mainly includes national roads and provincial roads; the secondary road includes roads below provincial; the arterial road is made up of the main roads inside urban. The Digital elevation model (30 m) is available from the United States Geological Survey (USGS) website (https://earthexplorer.usgs.gov, accessed on 28 June 2021). To simulate future LULC change, various types of data were processed correspondingly based on TerrSet and ArcGIS 10.8 software, and the results are shown in Figure 2. On the whole, the driving factors can be divided into natural class, including elevation, slope, annual average temperature, annual average precipitation, distance to rivers, soil type, and socio-economic class, including distance to highways, distance to railways, distance to railway stations, distance to governments, distance to arterial roads, distance to primary roads, distance to secondary roads, population density and GDP.

The flowchart of this study mainly includes three components (Figure 3). First, multiple types of the collected natural and socio-economic data were processed to obtain the driving factor dataset and expansion map that meet the experimental requirements. Second, the intensity and characteristics of LULC change in Hengyang were quantified by applying Intensity Analysis method and a new transition pattern. Third, the LULC demand under different scenarios by Markov Chain is forecasted, and then, the PLUS model is used to allocate that demand. The spatiotemporal change of LULC under different scenarios is compared and analyzed.

### 2.3. Intensity Analysis

Intensity Analysis method is a quantitative analysis method for analyzing category change [12]. It excavates the change information deeply based on the transfer matrix and obtains the different LULC patterns of the interval, category, and transition levels. The calculated intensity of LULC change at each level is compared with the corresponding uniform intensity (Figure 3). The interval level is to compare the total changes of different time intervals, and the category level compares the loss and gain of each LULC type in each time interval. The transition level compares the size and intensity of other categories’ transition to a specific category. Equations (1)–(6) describes the calculation method of Intensity Analysis [12] and the mathematical notation (Table 1).

Equations (1) and (2) respectively calculate the annual change rate at interval [*Y_t_*, *Y_t+1_*] and the annual uniform rate during the whole period [*Y*_1_, *Y*_T_]. If *U* = *S_t_*, the change at the interval level is stationary, meaning that all intervals were distributed uniformly during the whole period [*Y*_1_, *Y_T_*]. If *S_t_* > *U*, the annual change rate at interval [*Y_t_*, *Y_t_*_+1_] is fast; otherwise, it is slow.
(1)$$S_{t}=\frac{\sum_{j=1}^{J}\left[\left(\sum_{i=1}^{J}C_{tij}\right)-C_{tjj}\right]/\sum_{i=1}^{J}\sum_{j=1}^{J}C_{tij}}{\left(Y_{t+1}-Y_{t}\right)}100\%$$
(2)$$U=\frac{\sum_{t=1}^{T-1}\left\{\sum_{j=1}^{J}\left[\left(\sum_{i=1}^{J}C_{tij}\right)-C_{tjj}\right]\right\}/\sum_{t=1}^{T-1}\left[\left(Y_{t+1}-Y_{t}\right)\sum_{i=1}^{J}\sum_{j=1}^{J}C_{tij}\right]}{\left(Y_{T}-Y_{1}\right)}100\%$$

Equations (3) and (4) respectively calculate the annual gain intensity of category *j* and the annual loss intensity of category *i* at interval [*Y_t_*, *Y_t+_*_1_]. Equation (1) gives the uniform intensity for this category level analysis at interval [*Y_t_*, *Y_t+_*_1_]. If *G_tj_* > *S*, the gain of category *j* is active; otherwise, it is dormant. If *L_ti_* > *S*, the loss of category *i* is active; otherwise, it is dormant.
(3)$$G_{tj}=\frac{\left[\left(\sum_{i=1}^{J}C_{tij}\right)-C_{tjj}\right]/\left(Y_{t+1}-Y_{t}\right)}{\sum_{i=1}^{J}C_{tij}}100\%$$
(4)$$L_{ti}=\frac{\left[\left(\sum_{j=1}^{J}C_{tij}\right)-C_{tii}\right]/\left(Y_{t+1}-Y_{t}\right)}{\sum_{j=1}^{J}C_{tij}}100\%$$

Equation (5) calculates the annual transition intensity from a category non-n to category *n* for gaining category *n*. Equation (6) calculates the uniform intensity for gaining category *n* for transition level analysis. If *R_tin_* > *W_tn_*, the gaining of category *n* targets category *i*; otherwise, the gaining of category n avoids category *i*.
(5)$$R_{tin}=\frac{C_{tin}/\left(Y_{t+1}-Y_{t}\right)}{\sum_{j=1}^{J}C_{tij}}100\%$$
(6)$$W_{tn}=\frac{\left[\left(\sum_{i=1}^{J}C_{tin}\right)-C_{tnn}\right]/\left(Y_{t+1}-Y_{t}\right)}{\sum_{j=1}^{J}\left[\left(\sum_{i=1}^{J}C_{tij}\right)-C_{tnj}\right]}100\%\;$$

### 2.4. Transition Pattern

When more time point maps and LULC categories are involved, researchers use the transition level of Intensity Analysis to be tedious. For example, when the study involves LULC maps with *M* time points and *N* categories, it needs to implement (*M* − 1) × *N* times comparison of transition intensity to determine whether each transition shows stationary characteristics. To more intuitively and quickly identify the size, degree of targeting or avoiding, and characteristics of category transition, a novel transition pattern was designed in this study. Figure 4 describes the transition pattern and how to identify stationary transition. The temporal stationary mode of change is reflected in the case that the transition from category *i* to category *j* shows the same characteristic (target or avoid) at continuous intervals [12]. The rows and columns represent losses of a category and gains of a category, respectively. In addition, the deviation between the transition intensity and the corresponding uniform intensity represents how intensely the transition targets or avoids. The bubble size represents the size of the transition; that is, the area of the transition accounts for the proportion of the study area. The color of the bubbles indicates that the transition intensity deviates from the corresponding uniform intensity. If the transition intensity is greater than the corresponding uniform intensity, the bubble color is deeper red. Similarly, if the transition intensity is smaller than the corresponding uniform intensity, the bubble color is deeper blue. Readers can compare the color of bubbles horizontally to identify whether the transition is stationary during the whole period. Figure 4 shows that if the color is consistent, the transition is stationary.

### 2.5. Dynamic Simulation of LULC

#### 2.5.1. The PLUS Model

The PLUS model includes two modules, a rule mining framework based on the land expansion analysis strategy (LEAS) and a CA based on multitype random patch seeds [56]. Based on CA, the PLUS model combines random seed generation and a threshold-decreasing mechanism to simulate the change of patch level of multi-type land use [57]. The PLUS mechanism can well meet the purpose of this study and can better understand the LULC change process on driving forces. Its software can be downloaded free from the https://github.com/HPSCIL/Patch-level_Land_Use_Simulation_Model (accessed on 2 August 2021) [37]. First, we used the historical LULC in 2005/2015 and the dataset of driving factors. In addition, rivers, lakes and reservoirs, and other waters are difficult to lose under the strict implementation of government on the protection of river biodiversity, hence the need for inputting constraints of significant water protection in the PLUS model. The LEAS module applied a random forest algorithm to capture the influence of the factors on the expansion of LULC types and determined the development potential of different types of land in the study area and the contribution of each driving factor to various types of change. Based on the growth probability for each LULC type, taking 2005 as the initial time, the LULC spatial pattern of Hengyang in 2015 was simulated, and the accuracy was verified with the actual observation in 2015. If the accuracy meets the research requirements, taking 2015 as the initial time, the LULC spatial pattern of Hengyang under different scenarios (business as usual, economic development, and ecological protection) in 2025, 2035, and 2045 was projected by setting land demand based on the Markov Chain, conversion cost matrix and neighborhood weight of category (a reference to category gain of Intensity Analysis). Various scenarios are detailed in Table 2.

#### 2.5.2. Validation

To verify the projection ability of the PLUS model, this paper employed the total operating curve (TOC) and a set of statistical indicators. TOC was improved based on the relative operating curve (ROC). In the cross-contingency table, ROC represents the ability of the model to distinguish between pixels changing and pixels not changing for a category [58]. Compared with ROC, TOC can provide more useful information in adopting the same data and graphics space [59,60]. TOC software is available from https://lazygis.github.io/projects/TOCCurveGenerator (accessed on 30 October 2021) [61]. Making the TOC for each category is based on the expensed Boolean map and gain probability map. In this study, the figure of merit (FoM) was obtained by overlaying the actual observation map in 2015 and the simulation map in 2015. FoM is defined by Equation (7).
(7)$$\mathrm{FoM}=\frac{\left(Hits\right)100\%}{Misses+Hits+Wrong\;Hits+False\;Alarms}$$
where *Misses* represent error pixels due to reference change simulated as persistence; *Hits* represent correct pixels due to reference change simulated as change; *Wrong Hits* represent error pixels due to reference change simulated as the change to the wrong category; *False Alarms* represent error pixels due to the reference persistence simulated as change [62].

## 3. Results

### 3.1. Analysis of LULC Change Intensity

Figure 5 shows the size and annual change rate in each interval. If a bar stops before the uniform line, then the interval is slow. If a bar extends beyond the uniform line, then the interval is fast. The left side of Figure 5 shows that the size of the change during the first interval is greater than the size of the change during the second interval, while the right side of Figure 5 indicates the reason, which is that the duration of the first interval is greater than the duration of the second interval. The right side of Figure 5 shows that the annual change rate during the third interval is faster than the first interval and the second interval. The change rate is not perfectly stationary due to all bars on the right being unequal to the uniform line. Meanwhile, Figure 5 also indicates that the land is more affected by socioeconomic and human activities in Hengyang during the third interval, resulting in the change rate increasing by three times.

Figure 6 shows the size and intensity of gain and loss by category at three intervals. If a bar stops before the uniform line, then the category is dormant. If a bar extends beyond the uniform line, then the category is active. The annual change size on the left side of Figure 6 shows that there is a clear gradual increase in the gross gain of built-up land, with the annual change area increasing from 2.2 km^2^ (1980–1995) to 16.9 km^2^ (2005–2015), while there was an increasing trend in the gross loss of cultivated and forest lands. The gross gain and loss of bare land are relatively small. The change intensity of the right side of Figure 6 shows that bare and built-up lands are steadily active because the annual gain intensity and the annual loss intensity of bare land and the annual gain intensity of built-up land are greater than the uniform line. Conversely, the change of forest land is dormant because the change intensity of gain and loss of forest is smaller than the uniform line, while the change size of forest is relatively great, indicating that forest accounts for a large percentage of Hengyang.

Figure 7 shows the size and intensity of the transition from other categories to built-up land at three intervals. The left side of Figure 7 shows the annual transition size, and the right side shows the annual transition intensity. If a bar stops before the uniform line, then the transition avoids. If a bar extends beyond the uniform line, then the transition targets. The left side of Figure 7 shows that the largest contribution of the built-up land’s gain was cultivated, followed by forest land. The right side of Figure 7 shows that cultivated targets the gain of built-up but forest and water avoid it in the whole period.

### 3.2. Mode of LULC Change

Figure 8 shows the LULC change characteristic of mode in Hengyang from 1980 to 2015. The size and color of the bubble represent the transition size percentage of Hengyang and transition intensity deviation, respectively. Figure 8 presents various transition modes. The transition processes from cultivated to built-up, from forest to grass, from grass to forest, and from built-up to cultivated lands are in stationary targeting mode. The transition processes from cultivated to grass, from forest to cultivated, etc., are in stationary avoiding mode.

### 3.3. Validation for Simulation

To verify the simulation accuracy of the PLUS model, the LULC data of 2005 and 2015 and the driving factors dataset were used to obtain the growth probability of every category in the LEAS module. Then, taking 2005 as the initial time, the CARS module simulated the spatial pattern of LULC in 2015 under BAU scenarios. The simulated results were compared with the actual observations in 2015, and then the FoM and TOC were calculated. The closer the area under the curve (AUC) of the TOC value is to 1, the better the accuracy of the model. The results of FoM and overall accuracy are 4.6% and 96.2%, respectively. We input the category’s growth probability and simulated the expansion map for the category into TOC software to obtain the TOC curve of category gain. Figure 9 shows that the AUC values of TOC of each category are greater than 0.5 (the baseline value), demonstrating the satisfactory quality of each category’s calculated growth probability by the LEAS module of the PLUS model in Hengyang [63].

### 3.4. Multi-Scenario Projection Based on the PLUS Model

We input the parameters of predicting land demand and conversion cost matrix into the PLUS model. In addition, we applied the neighborhood weight parameter referring to the annual gain intensity of the category level from 2005 to 2015. Based on LULC data in 2015, the LULC spatiotemporal pattern would be predicted in Hengyang under BAU, ED, and EP scenarios in the CARS module of the PLUS model (Figure 10, Table 3). Zone 1 located in Hengyang County and Zone 2 located in Qidong County are magnified in Figure 10.

Figure 10 shows that the built-up expansion mainly occurred near the original place and on both sides of railways and roads, and the growth of built-up land is fastest under the ED scenario, while it was found that the westward expansion of Hengyang is obvious, and the city built-up land will cross the border with Hengyang County (Zone 1) in Figure 10d–f. The expansion of built-up land in Zone 2 is relatively slow compared to Zone 1. There may be two reasons for this: First, Zone 2 is mainly affected by terrain. Its northern part is mostly forest and at a higher elevation, which limits human activities and exploitation. Second, Zone 2 is far away from the city and is not sufficiently radiated and driven by urban growth in the city. Table 3 shows that water will increase and bare land will decrease, but the range of change is relatively small. In addition, cultivated, forest and grass lands will reduce. In the reducing rate, forest is largest, followed by cultivated and grass under the same scenarios, and ED is largest, followed by BAU and EP for the same category. For the rapid development in Hengyang, the reduction of forest, cultivated and grass lands can be alleviated under ecological protection policies and management. Since cultivated land is the main area around the urban area of Hengyang, it is difficult to avoid the built-up land occupation of cultivated land in regard to the urbanization and industrialization of Hengyang. When the problem of grain output decline is caused by cultivated reduction, the government needs to transform other types of land into cultivated land from farther away from urban land to achieve the “requisition–compensation balance” of cultivated land.

## 4. Discussion

### 4.1. Pattern and Process of LULC Change in Hengyang

In land change science, patterns and processes of LULC change are often discussed for the analysis of historical land change characteristics [20]. Intensity Analysis here shows that LULC change has undergone a complex process in Hengyang from 1980 to 2015, and among the three intervals, the 2005–2015 interval registered the largest annual change rate, which was three times that of the other two intervals, indicating the rapid change in this specific period. Another important finding is that the 1980–1995 and 1995–2005 intervals shared similar annual change (Figure 5), indicating that in the past 35 years, land change has accelerated since 2005 in Hengyang. This can be explained by the fact that Hengyang has taken reformational measures to answer to the “Rise of Central China” policy initiated by the Chinese Government in 2003, and urgent economic needs and regional resource development have significantly altered LULC in the area of study. Both have led to the rapid growth in land change size and intensity. With LULC structure statistics and a transition matrix, scientists will be able to identify land patterns and transition size [43,64]. However, scientists need to pay attention to what is driving category increases, decreases, and transition size. Figure 7 and Figure 8 show that the transition from cultivated and forest lands to built-up land was large. Compared with previous studies on Hengyang, category and transition levels of intensity analyses can better interpret the pattern and process phenomena. Figure 7 shows that the gain of built-up lands targeted cultivated but avoided forest lands. The transition from forest to built-up land was large since the original size of forest was large. The transition from cultivated to built-up land was large as well, due to the following two reasons: First, the original size of cultivated land was large. Second, built-up land stably gained targeted cultivated land (Figure 8). In addition, Figure 8 shows that the mutual transition between cultivated and water lands is stationary targeting as well, due to the following two reasons. First, water spreads across Hengyang, hence a greater possibility of being occupied in land change processes. Second, cultivated and water lands interact as a mosaic structure since rice farming in Hengyang depends on water. Therefore, the problem for future research is “how to quantify the neighborhood effects between land types”. Researchers should invent a land category adjacency matrix where each item value represents a certain adjacency length.

### 4.2. New Transition Pattern to Communicate Mode of LUCC

Compared with traditional methods such as transition matrix and dynamic degree, intensity analyses can reveal in-depth patterns and processes of LULC change in a top-to-bottom analysis [65]. For that, it has been applied to the study of geographic units of different scales and periods, providing a new quantitative research method for land change science. In the past, intensity analyses were often visualized by bar charts, where bar length represents category change size and intensity and is compared with a uniform line to identify land category change characteristics such as fast, slow, active, dormant, target, and avoid [20]. There are two defects in the traditional transition level. First, a large number of bar graphs are generated to express the size and intensity of different land categories at various intervals. If the transition level design here follows that of Aldwaik and Pontius Jr [12], 18 bar charts will be needed for three time intervals and six land categories, which is relatively complicated. Therefore, previous studies often chose critical land types for analysis [14,21]. For example, Quan et al. [20] only discussed three situations at the transition level: the gain of built, the gain of cultivated, and the gain of forest. Space limit has reduced scientists to such a situation that they cannot simply provide all transition information on one page. Second, it is inconvenient to compare the transition results of different time intervals in the transition level of the original Intensity Analysis, in which each time interval is represented by a single bar chart. Figure 6 in Fahad et al. [66] shows the gain intensity of four intervals and six land types with 16 bar charts, creating difficulty for temporal comparison to identify stationary characteristics.

Given the above defects, some scholars, based on Intensity Analysis, have proposed a few visualization solutions, e.g., a cross contingency table to show whether each transition targets or avoids [22] and a transition pattern to show transition size and the degree of targeting or avoiding [23,24,67]. However, since the transition pattern is based on time intervals, it fails to display LULC change characteristics and modes, e.g., stationary. In this study, we have offered a new transition pattern with the potential to improve Intensity Analysis methodology. This diagram shows all time intervals at the same time and can provide more information besides size and intensity. For example, in Figure 9, in each transition frame, we can determine stationary characteristics by checking if the color of bubbles is consistent in the horizontal direction. For example, different from Figure 11 in Xie et al., 2020 [24], Figure 9 shows that the size and intensity of transition under each interval are horizontally arranged to facilitate scientists in quickly identifying the temporal stability of the transition process, instead of showing the transition size and intensity by the interval. We thus recommend researchers to use this transition pattern. It can obtain full transition information, explore LULC change characteristics and modes in a more comprehensive way, and facilitate in comparing different regions so that they can discover similarities in land change modes, make clusters, and discuss the possibility of promoting a land policy that has been proven effective in a same-type land with similar LULC change modes. Thus, we have a question for future work: “How to establish a similarity coefficient to further cluster for multiple regions”.

## 5. Conclusions

(1) From 1980 to 2015, forest occupied the largest area, followed by cultivated land. The annual change rates of LULC are non-stationary between the three intervals. Before 2005, the change was slow. The annual change rate for the 2005–2015 interval was three times that of the other two.

(2) The gain of built-up and bare lands and the loss of cultivated land is stationary and active. The loss of forest is large but dormant, indicating that the large loss of forest is attributable to its initial size in the area of study. Cultivated and forest lands contribute significantly to built-up gains, but with different causes, namely, intensity and large initial size. The process of cultivated transition to built-up was stationary targeting mode, while the process of forest transition to grass was stationary avoiding mode.

(3) The PLUS model demonstrates high accuracy and applicability in the dynamic simulation of LULC change in Hengyang. Under the three scenarios, built-up land will expand along roads, while cultivated, forest, and grass lands will decrease. In the scenario of ecological protection, the decrease in forest, cultivated, and grass lands will be curbed, and the expansion of built-up land will be slowed down, to reconcile human–land conflicts. This will provide a reference for the sustainable development of LULC in Hengyang.

## Figures and Tables

**Figure 1 ijerph-19-08491-f001:**
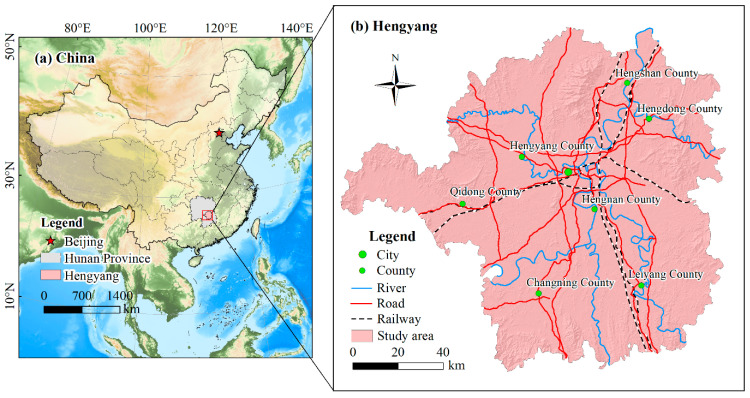
The geographical location of the study area.

**Figure 2 ijerph-19-08491-f002:**
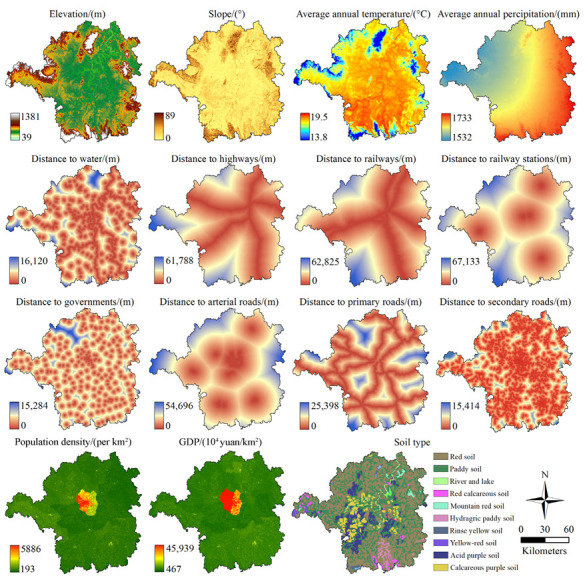
Spatial variables related to LULC change in Hengyang.

**Figure 3 ijerph-19-08491-f003:**
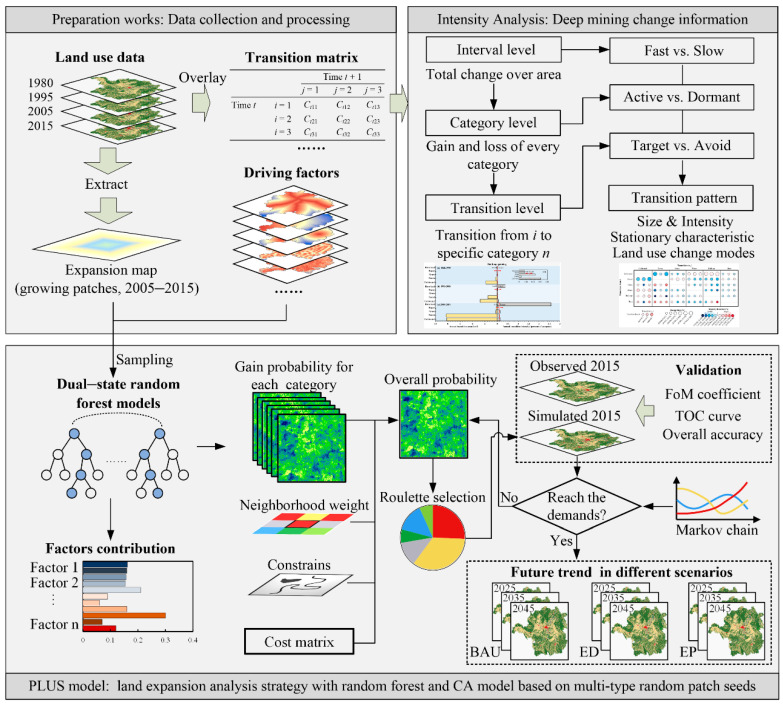
Research framework. PLUS: patch-generating land use simulation; BAU: business as usual scenario; ED: economic development scenario; EP: ecological protection scenario.

**Figure 4 ijerph-19-08491-f004:**
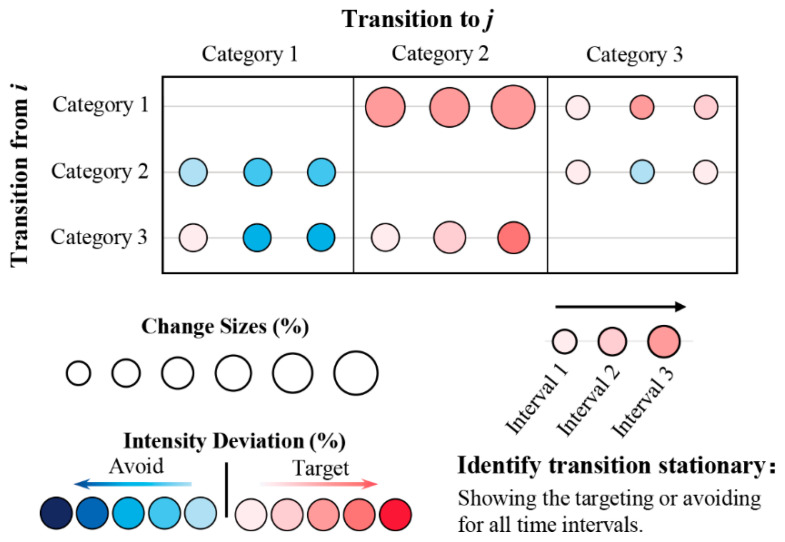
Schematic diagram of transition pattern.

**Figure 5 ijerph-19-08491-f005:**
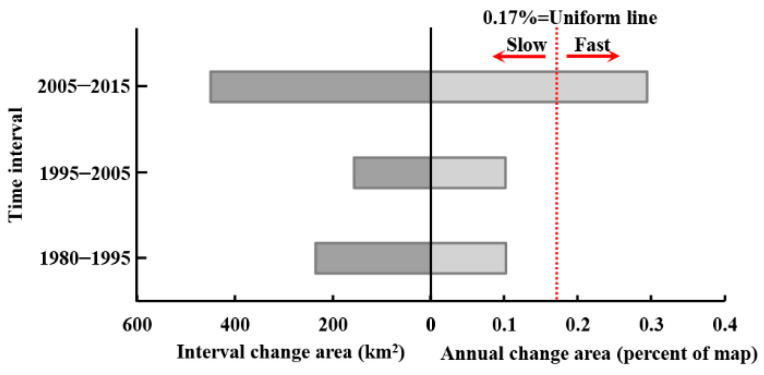
Size and annual change rate by interval from 1980–2015.

**Figure 6 ijerph-19-08491-f006:**
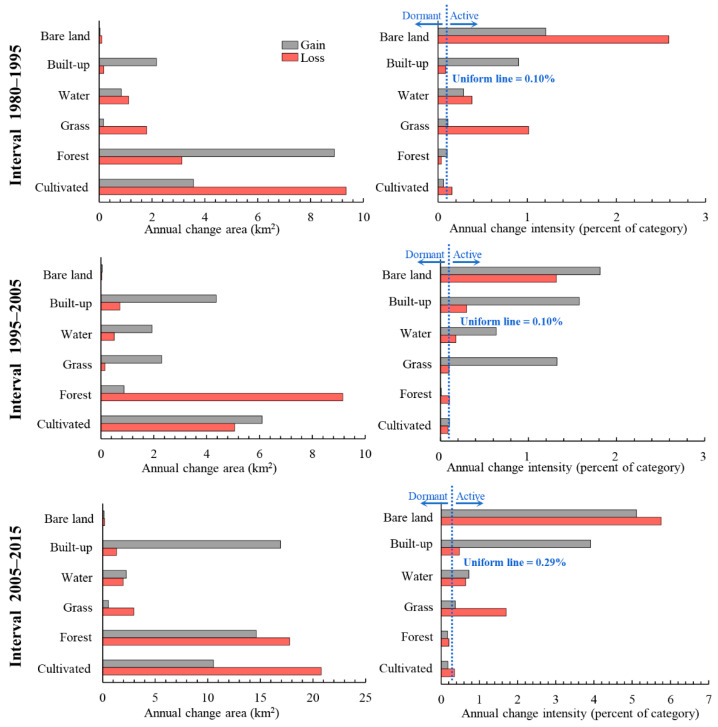
Annual size and intensity by category at three intervals.

**Figure 7 ijerph-19-08491-f007:**
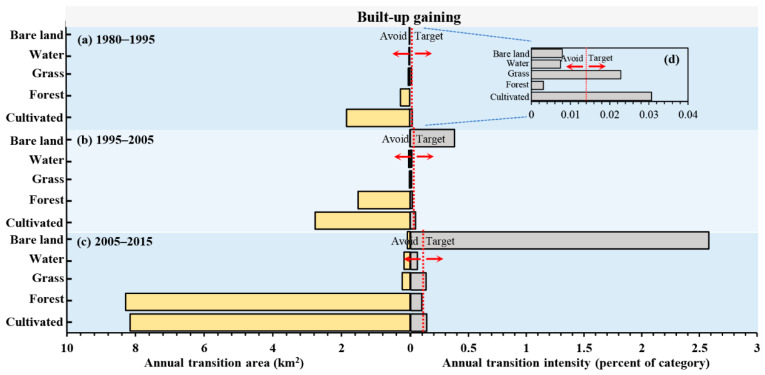
Annual transition size and intensity of the transition from other categories to built-up land at three intervals (**a**) 1980–1995, (**b**) 1995–2005, (**c**) 2005–2015); (**d**) magnified view of intensity from 1980–1995.

**Figure 8 ijerph-19-08491-f008:**
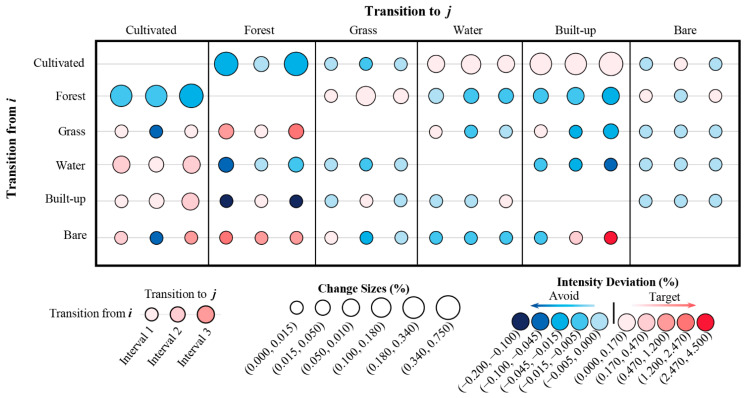
LULC change characteristic of mode in Hengyang from 1980–2015 (each bubble represents the transition from *i* to *j* at the interval; interval 1, interval 2 and interval 3 refer to the 1980–1995 period, 1995–2005 period and 2005–2015 period, respectively; intensity deviation = *Rtin* − *Wtn*).

**Figure 9 ijerph-19-08491-f009:**
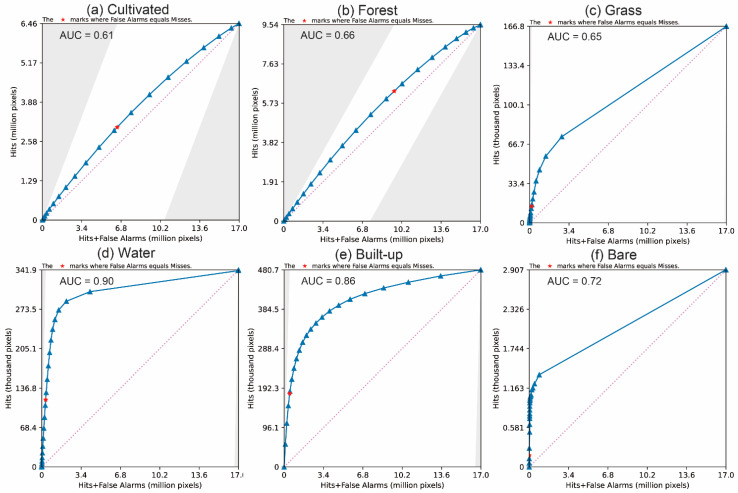
The TOC of simulation in 2015 by category: (**a**) cultivated, (**b**) forest, (**c**) grass, (**d**) water, (**e**) built-up, (**f**) bare.

**Figure 10 ijerph-19-08491-f010:**
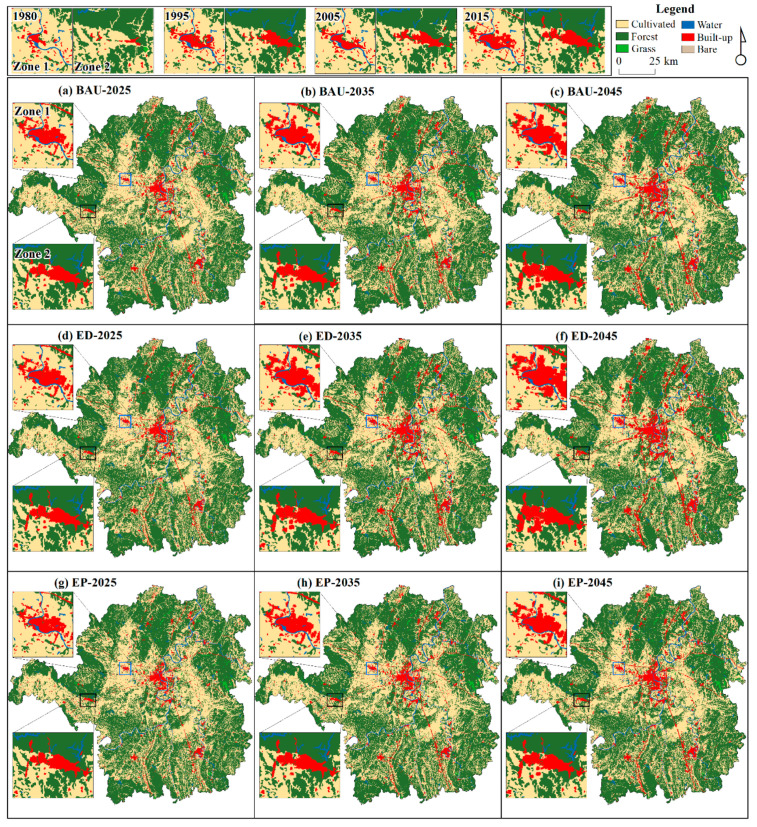
Projected pattern under a multi-scenario in 2025, 2035 and 2045 (BAU: business as usual scenario; ED: economic development scenario; EP: ecological protection scenario).

**Table 1 ijerph-19-08491-t001:** Mathematical notation of Intensity Analysis.

Symbol	Meaning
*T*	Number of time points
*Y_t_*	Year of time point *t*
*t*	Index of the initial time point of time interval [*Y_t_*, *Y_t_*_+1_], extent 1 to *T* − 1
*J*	Number of categories
*i*	Index of a category at the initial time point for a time interval
*j*	Index of a category at the final time point for a time interval
*C_tij_*	Number of pixels from category *i* at time *Y_t_* to category *j* at time *Y_t_*_+1_
*C_tji_*	Number of pixels from category *j* at time *Y_t_* to category *i* at time *Y_t_*_+1_
*S_t_*	Annual change rate at interval [*Y_t_*, *Y_t_*_+1_]
*U*	Uniform change rate at whole time extent [*Y*_1_, *Y_t_*_+1_]
*L_ti_*	Annual intensity of gross loss of category *i* at interval [*Y_t_*, *Y_t_*_+1_]
*R_tin_*	Annual transition intensity from category *i* to particular category *n* at interval [*Y_t_*, *Y_t_*_+1_], where *I ≠ n*
*W_tn_*	Uniform intensity from the non-n category to category *n* at interval [*Y_t_*, *Y_t+_*_1_]

**Table 2 ijerph-19-08491-t002:** Scenario setting based on the PLUS model.

Scenario Type	Description
Business as usual (BAU)	Based on the LULC change rate from 2005 to 2015 and historical natural and socioeconomic driving factors, this scenario uses the Markov Chain to predict the future land demand of various types, which is a demand parameter in the PLUS model. It is the basis for other scenarios.
Economic development (ED)	Socioeconomic backward is one of the major problems in underdeveloped regions. Therefore, with economic development, built-up land expansion is a prominent manifestation of backward regions. Based on BAU, this scenario modifies the transfer probability of the Markov Chain and increases the transfer probability from cultivated, grass, forest, and water to built-up by 50%, 30%, 10%, and 10%, respectively.
Ecological protection (EP)	With the overall protection of mountains, rivers, forests, farmland, lakes, and grasslands presented, the idea of ecological civilization has been deeply rooted in the hearts of the people in China. Based on BAU, since the built-up is more affected by human activities, the transfer probability from other types to built-up is reduced by 20% while the transfer probability of cultivated, forest, grass, and water to other types is reduced by 10% for protecting ecological land.

**Table 3 ijerph-19-08491-t003:** The PLUS model prediction results under different scenarios (unit: km^2^).

Category (Area in 2015)	Business as Usual			Economic Development			Ecological Protection		
	2025	2035	2045	2025	2035	2045	2025	2035	2045
Cultivated (5816.0)	5717.8	5631.9	5553.4	5678.5	5557.2	5447.7	5736.7	5667.8	5604.5
Forest (8588.6)	8531.9	8493.8	8453.6	8522.8	8475.5	8425.7	8544.0	8518.6	8491.4
Grass (150.1)	129.9	117.0	110.5	129.4	112.6	107.3	132.1	120.5	116.7
Water (307.7)	309.5	312.6	315.9	309.2	312.2	315.5	309.6	312.7	320.1
Built-up (432.6)	578.9	712.7	834.6	628.0	810.5	971.9	545.6	648.6	735.6
Bare (2.6)	2.4	2.4	2.3	2.4	2.3	2.2	2.3	2.2	2.1

## Data Availability

Not applicable.

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
