# Peer review of "Intensity Characteristics and Multi-Scenario Projection of Land Use and Land Cover Change in Hengyang, China"

_ijerph, 2022, doi:10.3390/ijerph19148491_

Round 1

Reviewer 1 Report

Authors have taken a good topic for projection of land use land cover in the Manuscript.  English Grammar needs to be checked with native English speaker before submission. 

I have few concern before the MS is accepted for publication. 

Literature review is not very carefully written as per scientific style, One should start with at least past methods, used for Land use projection, and land use in GIS as well as RS communities. 

Secondly, Authors have not considered the dimensions of remotely sensed images for this study, like spatial, spectral and temporal (for more information refer to Pandey, et al . (2021). Land use/land cover in view of earth observation: data sources, input dimensions, and classifiers—a review of the state of the art. Geocarto International36(9), 957-988. https://doi.org/10.1080/10106049.2019.1629647)

My minor comments-

abstract- line 22- "the loss of forest, cultivated and grass will be slow, while water will increase a little"  this sentence means loss of grass, but author wants to convey the loss of regions. kindly check and modify accordingly. 

Introduction- 

Include previous techniques for LULC , change and urban expansion analysis as suggested later on in this review comments. 

line 142- "LULC classification used for the first category is named as cultivated, forest, grass, water, built-up and bare" = Authors here means to say Level I. pls modify. and modify the sentences as well. 

Figure 2, elevation map, if there are no regions with less than 0 MSL , kindly change legend from 0 to show clear legend for map., 

Give space between equation 1 and 2 , they are closed enough and looks like one. 

Figure 8, categorized information should be very clear, i will suggest author to put number 1, 2 and 3 inside the circle of the figures too. 

Refer to the below articles for more information and contents. 

1. Khawaldah, H. A., Farhan, I., & Alzboun, N. M. (2020). Simulation and prediction of land use and land cover change using GIS, remote sensing and CA-Markov model. Global Journal of Environmental Science and Management6(2), 215-232.

2. Wang, J., & Maduako, I. N. (2018). Spatio-temporal urban growth dynamics of Lagos Metropolitan Region of Nigeria based on Hybrid methods for LULC modeling and prediction. European Journal of Remote Sensing51(1), 251-265. 

3. Karimi, F., Sultana, S., Babakan, A. S., & Suthaharan, S. (2021). Urban expansion modeling using an enhanced decision tree algorithm. GeoInformatica25(4), 715-731. 

4. Sharma, L., et al. (2012). Assessment of land consumption rate with urban dynamics change using geospatial techniques. Journal of Land Use Science7(2), 135-148. 

5. Zeng, C., Zhang, M., Cui, J., & He, S. (2015). Monitoring and modeling urban expansion—A spatially explicit and multi-scale perspective. Cities43, 92-103. 

Author Response

Dear editor Minerva Apetean and dear reviewers

Thank you for your letter and the reviewers’ comments concerning our manuscript entitled “Intensity Characteristics and Multi-scenario Projection of Land Use and Land Cover Change in Hengyang, China” (ijerph-1751149). Those comments are valuable and very helpful. We have read comments carefully and have made corrections. Based on the instructions provided in your letter, now we have uploaded the file of the revised manuscript. Our manuscript has marked up using the “Track Changes” function in Word 2016.

We would like to thank you for allowing us to resubmit a revised copy of the manuscript and we highly appreciate your time and consideration.

Manuscript ID: ijerph-1751149

Title: Intensity Characteristics and Multi-scenario Projection of Land Use and Land Cover Change in Hengyang, China

Your sincerely,

Authors

Reviewer #1:

I thank the referee for pointing out that the land use and land cover (LULC) projection is a good topic. It can help us to understand future probably trend of the LULC change. Thank you for your comment again. We really appreciate your efforts in reviewing our manuscript. We have revised the manuscript accordingly. Our point-by-point responses are detailed as follow.

Comment 1: English Grammar needs to be checked with native English speaker before submission.

Response: We apologize for the language problems in the original manuscript. The language presentation was improved with assistance from a native English speaker with appropriate research background.

Comment 2: Literature review is not very carefully written as per scientific style. One should start with at least past methods, used for land use projection, and land use in GIS as well as RS communities.

Response: We are grateful for the suggestion. To be more clearly and in accordance with the reviewer’s concerns, we have added more literature review regarding past method applied for LULC change and projection in GIS and RS. More detail was added on the Introduction section (page 1 line 39-43, page 2 line 77-93).

page 1 line 39-43: “Over the past decades, remote sensing (RS) and geographical information systems (GIS) have provided strong data sources and technical support for the monitoring and detecting of LULC change and urban expansion and sprawl (Pandey, P. C. et al. Land use/land cover in view of earth observation: data sources, input dimensions, and classifiers—a review of the state of the art. Geocarto International 2021, 36 (9), 957-988.). Compared to conventional ground surveys, RS has remarkably helped in the efficient acquisition and storage of earth surface data, which facilitates the quantification of long-time series LULC dynamics (Sharma, L. et al. Assessment of land consumption rate with urban dynamics change using geospatial techniques. Journal of Land Use Science 2012, 7 (2), 135-148.).”

page 2 line 77-93: “To comprehend probably trends in LULC, the spatiotemporal simulation of LULC is used as a powerful tool to analyze the consequence and the underlying driving force of regional human activities interacting with the natural environment [25]. Cellular Automata (CA) was first applied to geographical modeling in 1979 [26], mainly used in urban development studies at the outset [27, 28], which effectively relates urban expansion to socio-economic development through the defining transition rules. But the single model or method still was flawed despite its existing merits itself, hence the need for integrating modeling in LULC simulation. The coupling of CA and Markov Chain (MC) model has become a common method to simulate the LULC evolution and spatial distribution due to its prediction based on process state, strong parallel computing, and dynamic expression ability [25, 29]. A large number of scholars have conducted in-depth research in this field, and gradually developed a series of hybrid models and algorithms, such as the DT-based urban expansion model [30], Multi-criteria evaluation (MCE)-CA-MC [31], Logistic regression (LR)-CA-MC [32], SLEUTH model [33], Artificial neural network (ANN)-CA [34], MLP-CA-MC [35], CLUE-S model [36] and FLUS model [25], etc.”

Comment 3: Authors have not considered the dimensions of remotely sensed images for this study, like spatial, spectral and temporal.

Response: Thank you for this valuable feedback. In the references you provided, we found a lot of valuable information on remotely sensed images and LULC change. Based on this, we recognize that inputting dimension from remotely sensed images is important for a scientific study of land change. In response to the comments, we have added lines 149-157 to the Data Sources and Processing (2.2) Section, detailing the sources of remote sensing image data and how they were processed to obtain the land use and cover products in this study.

lines 149-157 to the 2.2 Section: “In this study, the LULC data in Hengyang in 1980, 1995, 2005, and 2015 (30 m resolution) are derived from the remote sensing monitoring dataset of land use and land cover in China, which can be provided by the Data Center for Resource and Environ-ment Sciences, Chinese Academy of Sciences (http://www.resdc.cn, accessed on 28 June 2021). The dataset was made by human-computer interaction visual interpretation based on Landsat TM/ETM+/OLI images, and its overall accuracy isn’t less than 90% (Liu, J. et al. Spatiotemporal characteristics, patterns, and causes of land-use changes in China since the late 1980s. Journal of Geographical Sciences 2014, 24 (2), 195-210.; Ning, J. et al. Spatiotemporal patterns and characteristics of land-use change in China during 2010–2015. Journal of Geographical Sciences 2018, 28 (5), 547-562.). According to the analysis for the study area, the LULC category in Level I was named as cultivated, forest, grass, water, built-up, and bare.”

Comment 4: Abstract- line 22- "the loss of forest, cultivated and grass will be slow, while water will increase a little" this sentence means loss of grass, but author wants to convey the loss of regions. kindly check and modify accordingly.

Response: Thank you for this feedback. It is improper to refer to "loss" as "area of loss" in the original manuscript. We want to convey that the loss rate of area in forest, cultivated and grass will slow down in the future. Therefore, the sentence be corrected as follows “The PLUS model confirmed that the forest, cultivated and grass will reduce and the rate of decrease will slow down in the future, while water will increase a little bit slightly” (see abstract-line21).

Comment 5: Include previous techniques for LULC change and urban expansion analysis as suggested later on in this review comments.

Response: We appreciate you for this kind recommendation. Land use and cover change has always been inextricably linked to urban expansion. Since the early 1980s, land use and cover in China has experienced unprecedented and dramatic changes in the context of rapid economic growth brought about by urbanization and industrialization (Wei, Y. D. & Ye, X. Urbanization, urban land expansion and environmental change in China. Stochastic Environmental Research and Risk Assessment 2014, 28(4), 757-765.). In other words, the urbanization or urban expansion in China has led to great changes in land use/cover structure. Therefore, in response to your comments, we have made the following revise and addition: (1) Introduction-line 40-“Over the past decades, remote sensing (RS) and geographical information system (GIS) have provided strong data sources and technical support for the monitoring and detecting of LULC change and urban expansion and sprawl (Pandey, P. C. et al. Land use/land cover in view of earth observation: data sources, input dimensions, and classifiers—a review of the state of the art. Geocarto International 2021, 36 (9), 957-988.).” (2) Introduction -line 59-For rich of application of Intensity Analysis method in urban expansion, we have added some references on page 2-line 60; (3) Tracing through your suggestions and the literature on land use projection, we found that projection models were mainly applied to urban sprawl in the early, which may be related to global economic development, population growth and the limitations of early geographical information technology. This is because early land use projection modelling existed that could only simulate the evolution of individual land types, such as urban sprawl, woodland degradation, etc. (Liu, X. et al. A future land use simulation model (FLUS) for simulating multiple land use scenarios by coupling human and natural effects. Landscape and Urban Planning 2017, 168, 94-116.). Therefore, we have made the following additions and modification: page 2-lines 78- “For comprehend probably trends in LULC, the spatiotemporal simulation of LULC is used as a powerful tool to analyze the consequence and its underlying driving force of regional human activities interacting with the natural environment [25]. Cellular Automata (CA) was first applied to geographical modelling in 1979 [26], mainly used in urban development studies at the outset [27, 28], which effectively relates urban expansion to socioeconomic development through the defining transition rules. But the single model or method still was flawed despite existing merits itself, hence the need for the integrating modeling in LULC simulation. Coupling of CA and Markov Chain (MC) model has become common method to simulate the LULC evolution and spatial distribution due to its prediction based on process state, strong parallel computing and dynamic expression ability (Khawaldah, H. A. et al. Simulation and prediction of land use and land cover change using GIS, remote sensing and CA-Markov model. Global Journal of Environmental Science and Management 2020, 6 (2), 215-232.).”

Comment 6: line 142- "LULC classification used for the first category is named as cultivated, forest, grass, water, built-up and bare" = Authors here means to say Level I. pls modify. and modify the sentences as well.

Response: We apologize for the language problems in this sentence. We have revised accordingly as follow “According analysis for study area, The LULC category in Level I is named as cultivated, forest, grass, water, built-up, bare. (2.2 Section-line 155)”.

Comment 7: Figure 2, elevation map, if there are no regions with less than 0 MSL, kindly change legend from 0 to show clear legend for map.

Response: We appreciate you for this kind recommendation. We have revised the information based on your comments and the Hengyang People's Government website as shown below.

Revised figure 2. Spatial variables related to the LULC change of the Hengyang.

Comment 8: Give space between equation 1 and 2, they are closed enough and looks like one.

Response: The layout here is not well thought out and may create an unclear reading for the reader. Therefore, we facilitate a more comfortable reading for the reader by setting the equation paragraph with 0.5 lines before and after the paragraph break.

Comment 9: Figure 8, categorized information should be very clear, i will suggest author to put number 1, 2 and 3 inside the circle of the figures too.

Response: Thank you for this comment. Figure 8 shows the proposed transition pattern for this study. It allows for a more intuitively and quickly identifiable stationary characteristic to detect some modes in land transition, such as the transition from cropland to building land as a stationary targeting transition mode. The category information in Figure 8 should be clear enough to allow the reader easily distinguish and find the meaning of each circle. By means of rows and columns, each circle corresponds to which land category is transfered, while the columns based on one category determine to which column this bubble belongs to, and thus to what time interval it belongs.

The addition of the numbers 1, 2, 3 to each of the circles may result in the intensity differences not being perceived. We have therefore made changes as shown in the figure 4 and figure 8 below.

Revised figure 4. Schematic diagram of transition pattern.

Revised figure 8. LULC change characteristic of mode in Hengyang during 1980-2015 (each bubble represents the transition from i to j at the interval; interval 1, interval 2 and interval 3 refer to 1980-1995 period, 1995-2005 period and 2005-2015 period respectively; intensity deviation= Rtin-Wtn)

Reviewer 2 Report

The article is interesting and clearly presented but I have some minor remarks that I think are critical to make the article easy and comfortable to read:

1. Figure 1 should be placed in "2.1. Study area" not in "2.2. Data source and processing".

2. Lines 142, 146, 149, 223 and 256 - it would be better to put the relevant links in the list of references, along with the date of access, and in the main text only to refer to specific items from references

3. Figure 2 should be placed in  "2.2. Data source and processing" not in "2.3. Intensity Aanalysis".

4. Figure 2 - it seems that for most maps on Figure 2 the color scale has a too sharp transition from red to yellow and further to blue, which gives the effect of a sharp borders between areas, while the values are more linearly distributed

5. Figure 2 - there is a lack of legend for last map "Soil type". 

6. Equations from (1) to (7) should be separated by blank lines to make them easier to read

7. Table 1 - try not to divide tables onto two separate pages.

8. Chapter "2.4. Transition pattern" should not begin with Figure 4. Therefore Figure 4 should be placed after first paragraph of the chapter.

9. Line 204 - start the figure caption from capital letter.

10. Figure 6 should be placed between present lines 293 and 294.

11. Figure 9 should be placed in "3.3. Validation for simulation" not in "3.2 Mode of LULC change".

12. Figure 10 should be placed in "3.4. Multi-scenario projection based on the PLUS model" not in "3.3. Validation for simulation". Please place Figure 10 after present line 340.

13. Line 340, 344 and 356 - there should be "Table 3" not "Table 2".

14. Line 344 - there should be "Figure 10d-f" not "Figure 11d-f".

In spite of proper design of research I also have some doubths about validation of simulation:

15. Line 326 - explain what "TOC closer to 1" is supposed to mean. Didn't the authors mean "AUC of TOC closer to 1"?

16. Line 327 - explain why FoM = 0.046 is good when the authors say that "The closer the FoM and TOC value are to 1, the better the simulation accuracy is." Shouldn't FoM be 0.046 * 100% = 4.6% and why is it good value to proof the accuracy of predictions?

17. Is the FoM calculated for all categories together? If it is, then why do not make TOC for all categories together?

18. Why "area under curve (AUC) values of TOC of each category both are greater than 0.65, indicating high accuracy of simulation"? Three of AUC are close to 0.65, two are high (about 0.9) and one in the middle. Maybe TOC of all categories togheter will show something more comprehensive and transparent?

Best regards.

Author Response

Reviewer #2:

I thank the referee for pointing out that quantifying change of land use and land cover (LULC) is interesting in our study. The projection by geographical modeling can help us to understand future probably trend of the LULC change. Thank you for your summary again. We really appreciate your efforts in reviewing our manuscript. We have revised the manuscript accordingly. Our point-by-point responses (red font) to your comments are detailed below.

Comment 1: Figure 1 should be placed in "2.1. Study area" not in "2.2. Data source and processing".

Response: Thank you for this valuable comment. Figure 1 have been placed in 2.1 Section. In order to make it clear for readers, we have modified Figure 1 by removing the four small maps on land use from the original Figure 1 due to the difficulty of distinguishing map differences and considerations for one page space.

Revised figure 1. The geographical location of study area.

Comment 2: Lines 142, 146, 149, 223 and 256 - it would be better to put the relevant links in the list of references, along with the date of access, and in the main text only to refer to specific items from references

Response: Thank you for this suggestion. We have added acquisition times and more detailed descriptions of the data in lines-142, 146, 149, 223 and 256 according to comments. We also some sentences in 2.2 Section Data source and processing as follow “In this study, the LULC data in Hengyang in 1980, 1995, 2005 and 2015 (30 m resolution) are derived from remote sensing monitoring dataset of land use and land cover in China, which can be provided by Data Center for Resource and Environment Sciences, Chinese Academy of Sciences (http://www.resdc.cn, access on 28 June 2021). The dataset was made by human-computer interaction visual interpretation based on Landsat TM/ETM+/OLI images, and its overall accuracy isn’t less than 90% (Liu, J. et al. Spatiotemporal characteristics, patterns, and causes of land-use changes in China since the late 1980s. Journal of Geographical Sciences 2014, 24 (2), 195-210. And Ning, J. et al. Spatiotemporal patterns and characteristics of land-use change in China during 2010–2015. Journal of Geographical Sciences 2018, 28 (5), 547-562.). According analysis for study area, The Level I of LULC category is named as cultivated, forest, grass, water, built-up, bare.”

Comment 3: Figure 2 should be placed in "2.2. Data source and processing" not in "2.3. Intensity Analysis".

Response: Thank you for this recommendation. Figure 2 have been placed in 2.2 Section.

Comment 4: Figure 2 - it seems that for most maps on Figure 2 the color scale has a too sharp transition from red to yellow and further to blue, which gives the effect of a sharp borders between areas, while the values are more linearly distributed.

Response: Thank you for this valuable comments. Based on your comments, we verify the data visualization issues in the ArcGIS 10.8. Some of maps where sharp borders were done based on the spatial analysis tool-Euclidean Distance in ArcGIS. The visualization results based on the distance tool are due to the fact that displaying such a large area of Hengyang on a small map will cause the colors to be displayed not very smoothly. Such a situation also occurs in other literatures (Figure 2 in Han, J. et al. How to Account for Changes in Carbon Storage from Coal Mining and Reclamation in Eastern China? Taking Yanzhou Coalfield as an Example to Simulate and Estimate. Remote Sensing 2022, 14 (9), 2014-2034.; Figure 3 in Li, C. et al. How Will Rwandan Land Use/Land Cover Change under High Population Pressure and Changing Climate? Applied Sciences 2021, 11 (12), 5376-5395.).

Figure 2 in Han, J. et al. (2022)

Figure 3 in Li, C. et al. (2021)

Comment 5: Figure 2 - there is a lack of legend for last map "Soil type".

Response: We have revised for soil type in the Figure 2 according to your feedback.

The soil type map of Revised figure 2. Spatial variables related to the LULC change of the Hengyang.

Comment 6: Equations from (1) to (7) should be separated by blank lines to make them easier to read.

Response: The layout here is not well thought out and may create an unclear reading for the reader. We have therefore separated the adjacent equations in equations (1)-(7) throughout the text by leaving a blank line to facilitate a more comfortable reading for readers.

Comment 7: Table 1 - try not to divide tables onto two separate pages.

Response: Thank you for this helpful suggestion. We will be sure to keep every table in this article from being displayed across pages before the manuscript is published. We will ask the editor to help us to solve the problem.

Comment 8: Chapter "2.4. Transition pattern" should not begin with Figure 4. Therefore Figure 4 should be placed after first paragraph of the chapter.

Response: For a scientific manuscript, any figure is usually placed after the first paragraph of the text, rather than with the figure before the first paragraph of text. We will certainly follow your comment to place Figure 4 after the text. Thank you for this suggestion.

Comment 9: Line 204 - start the figure caption from capital letter.

Response: We have modified this according to your valuable suggestion, which can be seen in Figure 4 in Section 2.4 “Transition Pattern”.

Comment 10: Figure 6 should be placed between present lines 293 and 294.

Response: Figure 6 shows the results regarding the category level. We have placed Figure 6 after the second paragraph of 3.1 Section “Analysis of LULC change intensity” according to your reminder.

Comment 11: Figure 9 should be placed in "3.3. Validation for simulation" not in "3.2 Mode of LULC change".

Response: We have revised this according to your valuable reminder. The figure 9 have been placed in 3.3 Section “Validation for simulation”.

Comment 12: Figure 10 should be placed in "3.4. Multi-scenario projection based on the PLUS model" not in "3.3. Validation for simulation". Please place Figure 10 after present line 340.

Response: As there is a limit to the spatial difference between the 9 maps in multi-scenario that be represented on one page, we have modified Figure 10 as appropriate. More spatial information was shown in Figure 10 to help readers get comfortable with comparison between multiple maps.

Revised figure 10. Projected pattern under multi-scenario in 2025, 2035 and 2045 (BAU: business as usual scenario; ED: economic development scenario; EP: ecological protection scenario).

Comment 13: Line 340, 344 and 356 - there should be "Table 3" not "Table 2"

Response: We have revised this Table’s caption number according to your valuable reminder.

Comment 14: Line 344 - there should be "Figure 10d-f" not "Figure 11d-f".

Response: Thank you for this helpful recommendation. We have revised this mistake (line-366).

Comment 15: Line 326 - explain what "TOC closer to 1" is supposed to mean. Didn't the authors mean "AUC of TOC closer to 1"?

Response: Thank you very much for your comment. based on improvement for ROC, the TOC was proposed to examine the ability of allocating, such as“non-built to built”in urban expansion simulation. The TOC can validate the predictive power of the extrapolation (Shafizadeh-Moghadam, Hossein. et al. Integrating a Forward Feature Selection algorithm, Random Forest, and Cellular Automata to extrapolate urban growth in the Tehran-Karaj Region of Iran. Computers, Environment and Urban Systems 2021, 87.). This study is in essence trying to express the AUC value of TOC and this description have been revised in 3.3 Section “Validation for simulation”. The closer the AUC value is to 1, the more accurate the model is. In other words, if the AUC value would be 0.5 or less than 0.5, the model is inaccurate.

Comment 16: Line 327 - explain why FoM = 0.046 is good when the authors say that "The closer the FoM and TOC value are to 1, the better the simulation accuracy is." Shouldn't FoM be 0.046 * 100% = 4.6% and why is it good value to proof the accuracy of predictions?

Response: Thank you for your feedback. Firstly, we apologize for the statement that "the closer the FoM accuracy is to 1, the better it is", as it suggests potentially that the FoM can easily approach 1. In fact, many studies have shown that FoM coefficients as high as 0.2 are indicative of very good model performance (Liang, X. et al. Understanding the drivers of sustainable land expansion using a patch-generating land use simulation (PLUS) model: A case study in Wuhan, China. Computers, Environment and Urban Systems 2021, 85.; Liu, X. et al. A future land use simulation model (FLUS) for simulating multiple land use scenarios by coupling human and natural effects. Landscape and Urban Planning 2017, 168, 94-116.). Secondly, The FoM is an accuracy indicator for spatial prediction proposed by Pontius and gradually enriched (Varga, O. G. & Pontius Jr, R. G. et al. Intensity Analysis and the Figure of Merit’s components for assessment of a Cellular Automata–Markov simulation model. Ecological Indicators 2019, 101, 933-942.). Based on 13 study cases from nine popular models, one huge finding is that applications that have larger amounts of observed net change in the reference maps tend to have larger predictive accuracies as measured by the figure of merit. Thus, due to the small net change in Hengyang observations, the FoM value is 4.6%. Finally, we would also like to explain why the Kappa coefficient was not used as a test in this study. This is because some literatures suggest that the Kappa coefficient mainly reflects the chance consistency of the data and is not a good indicator for monitoring the predictive power of the model (Pontius Jr, R. G. et al. Death to Kappa: birth of quantity disagreement and allocation disagreement for accuracy assessment. International Journal of Remote Sensing 2011, 32 (15), 4407-4429.; Foody, G. M. Explaining the unsuitability of the kappa coefficient in the assessment and comparison of the accuracy of thematic maps obtained by image classification. Remote Sensing of Environment 2020, 239.).

Comment 17: Is the FoM calculated for all categories together? If it is, then why do not make TOC for all categories together?

Response: Thank you for this helpful comment. Firstly, the FoM was calculated for all category together, more about the FoM in Varga, O. G. & Pontius Jr, R. G. et al. (2019). Secondly, the TOC allocate the ability to test the model to allocate targeted category and non-targeted categories. For example, the ability of the model to allocate built versus non-built when the model simulates urban expansion. As such, it can be used to compare the extrapolation capabilities of different models or algorithms in urban sprawl analysis (Kamusoko, C. et al. Simulating Urban Growth Using a Random Forest-Cellular Automata (RF-CA) Model. ISPRS International Journal of Geo-Information 2015, 4 (2), 447-470.). In other words, TOC is often used to validate the ability of individual category to allocate, or to validate each simulated category individually to give researchers an idea of the model’s ability to spatially allocate each category (Shafizadeh-Moghadam, Hossein et al. Coupling machine learning, tree-based and statistical models with cellular automata to simulate urban growth. Computers, Environment and Urban Systems 2017, 64(1), 297–308.; Chen, Y. Y. et al. Tele-connecting China's future urban growth to impacts on ecosystem services under the shared socioeconomic pathways. Sci Total Environ 2019, 652, 765-779.). There is insufficient evidence to suggest that TOC can examine well for all categories (for overall land use change). Therefore, TOC has not been used to examine this situation in this study.

Comment 18: Why "area under curve (AUC) values of TOC of each category both are greater than 0.65, indicating high accuracy of simulation"? Three of AUC are close to 0.65, two are high (about 0.9) and one in the middle. Maybe TOC of all categories together will show something more comprehensive and transparent?

Response: Thank the reviewer very much for the valuable comments. We have revised the section 3.3 Validation of simulation. Firstly, we agree that the reviewer's comment that "TOC values are greater than 0.65" is not appropriate. This is because studies of the TOC have shown that it is the baseline of the TOC that is important as an indicator of whether the model is satisfactory or unsatisfactory (Chen, Y. et al. Tele-connecting China's future urban growth to impacts on ecosystem services under the shared socioeconomic pathways. Sci Total Environ 2019, 652, 765-779.). In addition, we have added the overall accuracy indicator in section 3.3 to measure the performance of the model from the prospective of multiple evaluation indicators. Finally, we are in agreement with the reviewer regarding the possibility that TOC may be more comprehensive for all categories. However, the PLUS model’s LEAS module based on the dual-state random forest approach is well able to mine the non-linear relationship between category growth and the driving factors, and determine the extent to which each factor contributes to each category, for example, assuming that the impact factor-temperature does not contribute significantly to category change in study area, then the LEAS module's output of category growth probability also considers the extent to which this factor affects its calculation. The PLUS model is able to estimate the growth probability of each land category with a combination of natural environmental and socioeconomic factors. Based on the results, the conversion cost matrix, domain impacts, and constraints are integrated to determine the future land use allocation rules (Liang, X. et al. Understanding the drivers of sustainable land expansion using a patch-generating land use simulation (PLUS) model: A case study in Wuhan, China. Computers, Environment and Urban Systems 2021, 85.). The PLUS model is currently able to provide outputs under the LEAS module: the degree of factor contribution and the map of the growth probability for each category. Therefore, the TOC is not yet well accomplished for all categories based on the PLUS model due to the unavailability of suitable TOC platform inputs.

Reviewer 3 Report

The article tries to present the issues of long-term changes in land use in an interesting way, together with an attempt to project these changes in the future.
The introduction was properly written, although too focused on the Chinese territory. This part of the article (and, to a lesser extent, also the discussion) could strongly refer to the results of world research on similar issues, especially long-term methods of forecasting land use changes.
Data sources well described. It seems to me (Figure 1) that it makes no sense to use four small maps showing land use from different periods, it is difficult to notice any differences in this way. Perhaps it is useful to prepare four high-resolution figures as attachments.
The data presented in Figure 2 are disturbing and require checks (or explanations). Is the study based on good data? Are there heights up to -77 m in the studied area? Are the rivers point-like rather than linear? An analysis of aerial photos shows that there are lakes, not just rivers. In this case, the only problem is the map description and this case can be justified. However, it is similar with the map showing "arterial roads", where also the spatial analysis is "point". What distance was determined? If for entry points on highways then it's okay. However, the same problem applies to "Secondary roads", the lowest category of roads and the continuity of this layer is certainly maintained. This study shows many "point" elements. Only the analysis of "Primary roads" seems correct. Perhaps it is worth presenting maps with data that were the basis for the analysis of distance zones.

Figure 10. For the reader, this is 9 identical maps. I hope that the final version of the research will contain high-resolution data.

All in all, the article reads well and presents interesting and new information. However, the problem with the data that was used for the analysis (mainly the "secondary roads)" requires an absolute explanation. Potential errors at this stage affect the results of the entire study.

Author Response

Reviewer #3:

I thank the referee for pointing out that quantifying change of land use and land cover (LULC) is interesting in our study. The projection by geographical modeling can help us to understand future probably trend of the LULC change. Thank you for your summary again. We really appreciate your efforts in reviewing our manuscript. We have revised the manuscript accordingly. Our point-by-point responses to your comments are detailed below.

Comment 1: It seems to me (Figure 1) that it makes no sense to use four small maps showing land use from different periods, it is difficult to notice any differences in this way.

Response: Thank you for this helpful comment. After careful consideration, we removed the four small land use maps from Figure 1. We have also optimized the visualization of Figure 1 for more comfortable reading.

Revised figure 1. The geographical location of study area.

Comment 2: The data presented in Figure 2 are disturbing and require checks (or explanations). Is the study based on good data?

Response: Thank you for this valuable feedback. We have revised the Section 2.2-Data source and processing to detail the materials and its processing.

Table. Summary of data sources in this manuscript

Data

Data sources

DEM

The Unites States Geological Survey (USGS) website (https://earthexplorer.usgs.gov, access on 28 June 2021).

Traffic road data

(multi-level highway networks; high-speed railways, railways; highways; governments; and other vector sources )

The non-secret vector map data product (1: 1 million scale). This product is provided by the National Geomatics Center of China (https://www.webmap.cn, access on 28 October 2021). Note: The Ministry of Natural Resources is the competent authority. The data is not secret and can therefore be downloaded on request free of charge.

The GDP, population grid data;

Annual precipitation and temperature distribution of 1 km resolution data;

Soil type data

The Data Center for Resource and Environment Sciences, Chinese Academy of Sciences (http://www.resdc.cn, access on 19 October 2021).

Land use and land cover

The remote sensing monitoring dataset of land use and land cover in China, which can be provided by Data Center for Resource and Environment Sciences, Chinese Academy of Sciences (http://www.resdc.cn, access on 28 June 2021).

Comment 3: Are there heights up to -77 m in the studied area?

Response: We appreciate you for this kind recommendation. We have revised the information based on your comments and the Hengyang People's Government website as shown below.

The DEM part of revised figure 2. Spatial variables related to the LULC change of the Hengyang.

Comment 4: Are the rivers point-like rather than linear? An analysis of aerial photos shows that there are lakes, not just rivers. In this case, the only problem is the map description and this case can be justified.

Response: Thank you for this valuable comment. We are wrong in this caption. We apologize for the confusion we have caused you in reading this. In fact, we were trying to convey "distance to water" rather than "distance to rivers". We have corrected this as follows.

The water part of revised figure 2. Spatial variables related to the LULC change of the Hengyang.

Comment 5: However, it is similar with the map showing "arterial roads", where also the spatial analysis is "point". What distance was determined? If for entry points on highways then it's okay.

Response: Thank the reviewer for these helpful comments. This is problem related to map scale. If the map is on a larger scale, the distance to this road may look like "lines"; If the map is on a smaller scale, the distance to this road may look like "points". The arterial road is made up of the main roads inside urban in this study (See page 2, line-166). The arterial road is the backbone of the city's road network and is the main traffic route linking all major subdivisions of the city. The roads themselves are relatively close together compared to the primary roads, so the distance maps for these roads seem to be easily grouped together ("points") at a small scale.

Hengyang Transportation Road Network excluding railways and high-speed railways (https://www.webmap.cn)

Comment 6: However, the same problem applies to "Secondary roads", the lowest category of roads and the continuity of this layer is certainly maintained. This study shows many "point" elements.

Response: Thank you for your valuable comments. For this situation, we think there are two reasons. Firstly, this a problem involving map scale, which is similar to Comment 5. Secondly, it is related to the processing of our data. The steps of its handling are as follows: (1) according to the attributes of the above traffic road data, we selected out highways (which require charges in China); (2) we selected out arterial roads, which refer to major roads inside urban; (3) we selected out primary roads, including national roads and provincial roads (which are free in China); (4) the remaining roads were set as secondary roads, which mainly include county roads and major town roads. As a result, secondary roads can easily be shown as "points" at a very small scale due to the relative dispersion and discontinuity of these roads.

Comment 7: Figure 10. For the reader, this is 9 identical maps. I hope that the final version of the research will contain high-resolution data.

Response: Thank you for your comments. We agree with the suggestion, and have modified Figure 10 as follows, to make the reader to see from the Figure 10 that the nine maps are not identical and to detect spatial differences. The Zone 1 located in Hengyang County and the Zone 2 located in Qidong County are zoomed in Figure 10.

Revised figure 10. Projected pattern under multi-scenario in 2025, 2035 and 2045 (BAU: business as usual scenario; ED: economic development scenario; EP: ecological protection scenario).

Round 2

Reviewer 1 Report

Authors have incorporated all suggested contents and MS now in scientific writing style. It may be now accepted for publication. 

Reviewer 2 Report

The Authors have dispelled all my doubts, so I have no further objections.

Best regards.

Reviewer 3 Report

The authors did a lot of good work on the previous version of the manuscript. The current version has been significantly improved and in my opinion is suitable for publication.